# Silane- and peroxide-free hydrogen atom transfer hydrogenation using ascorbic acid and cobalt-photoredox dual catalysis

Yuji Kamei[1], Yusuke Seino[1], Yuto Yamaguchi[1], Tatsuhiko Yoshino [1], Satoshi Maeda [2,3,4], Masahiro Kojima [1✉] & Shigeki Matsunaga [1,5✉]

Hydrogen atom transfer (HAT) hydrogenation has recently emerged as an indispensable method for the chemoselective reduction of unactivated alkenes. However, the hitherto reported systems basically require stoichiometric amounts of silanes and peroxides, which prevents wider applications, especially with respect to sustainability and safety concerns. Herein, we report a silane- and peroxide-free HAT hydrogenation using a combined cobalt/photoredox catalysis and ascorbic acid (vitamin C) as a sole stoichiometric reactant. A cobalt salophen complex is identified as the optimal cocatalyst for this environmentally benign HAT hydrogenation in aqueous media, which exhibits high functional-group tolerance. In addition to its applicability in the late-stage hydrogenation of amino-acid derivatives and drug molecules, this method offers unique advantage in direct transformation of unprotected sugar derivatives and allows the HAT hydrogenation of unprotected C-glycoside in higher yield compared to previously reported HAT hydrogenation protocols. The proposed mechanism is supported by experimental and theoretical studies.

[1] Faculty of Pharmaceutical Sciences, Hokkaido University, Sapporo 060-0812, Japan. [2] Institute for Chemical Reaction Design and Discovery (WPI-ICReDD), Hokkaido University, Sapporo 001-0021, Japan. [3] Faculty of Science, Hokkaido University, Sapporo 060-0810, Japan. [4] JST, ERATO Maeda Artificial Intelligence for Chemical Reaction Design and Discovery Project, Sapporo 060-0810, Japan. [5] Global Station for Biosurfaces and Drug Discovery, Hokkaido University, Sapporo 060-0812, Japan. ✉email: m-kojima@pharm.hokudai.ac.jp; smatsuna@pharm.hokudai.ac.jp

Pioneered by Drago and Mukaiyama in the late 1980s, Markovnikov-selective hydrofunctionalizations of alkenes via hydrogen atom transfer (HAT) have evolved into versatile tools in the synthesis of complex molecules[1,2]. Among the selected recent contributions by Carreira[3–7], Boger[8,9], Baran[10–14], Shenvi[15–20], Herzon[21–23], and others[24–28], the HAT hydrogenation of alkenes was introduced independently by Shenvi and Herzon and has since been established as a complementary method for commonly used noble-metal-catalyzed hydrogenation reactions. Shenvi has reported Mn- and Co-catalyzed protocols that use phenylsilane or isopropoxy(phenyl)silane reductants in the presence of tert-butyl hydroperoxide, which tolerates benzyl group and C(sp²)–halogen bonds[29,30]. In addition, thermodynamically controlled stereoselectivity is characteristic of these transformations and supports the proposed carbon-based radical intermediates (Fig. 1a). Herzon has reported Co-catalyzed hydrogenation protocols using triethylsilane, tert-butyl hydroperoxide, and 1,4-cyclohexadiene[31–33]. This method possesses high functional-group tolerance and was studied in detail for the hydrogenation of haloalkenes and the selective hydrogenation of substituted alkenes (Fig. 1b). While these systems have been increasingly appreciated in natural-product synthesis[34,35], stoichiometric amounts of organosilanes and peroxides are required in order to achieve a wide substrate scope, which remains problematic in terms of sustainability and safety concerns. In this context, Norton's cobaloxime-catalyzed protocol allowed hydrogenation of acrylamides using only molecular hydrogen as the sole stoichiometric reactant, and should thus be considered as an insightful example of the silane- and peroxide-free HAT

**a Silane, peroxide, & Mn or Co catalyst (Shenvi)**

Unique FG tolerance
Thermodynamic stereoselectivity

**b Silane, peroxide, H• donor, & Co catalyst (Herzon)**

Unique FG tolerance
Selective for substituted alkenes

**c H₂ & Co catalyst (Norton)**

Silane- and peroxide-free
Limited to electron-deficient alkenes

**d Ascorbic acid & Co/photoredox (This work)**

Unique FG tolerance
Silane- and peroxide-free
Suitable for hydrophilic compounds

**Fig. 1 Catalytic HAT hydrogenation of alkenes. a** Shenvi's method. **b** Herzon's method. **c** Norton's method. **d** This work; dpm 2,2,6,6-tetramethyl-3,5-heptanedionato, FG functional group, acac 2,4-pentanedionato, Cy cyclohexyl, DTBMP 2,6-di-tert-butyl-4-methylpyridine, dmg dimethylglyoximato, EWG electron-withdrawing group.

**Fig. 2 Proposed mechanism for the HAT hydrogenation using ascorbic acid and a combined cobalt/photoredox catalytic system. a** Hypothetical catalytic cycle of the silane- and peroxide-free HAT hydrogenation. **b** Disproportionation of ascorbic acid radicals; PC photoredox catalyst.

hydrogenation of alkenes (Fig. 1c)[36]. Norton's studies revealed fundamental physical parameters associated with HAT hydrogenations, albeit that the substrates hydrogenated in high yield were limited to alkenes activated with electron-withdrawing groups. During the review of this manuscript, Kattamuri and West reported iron and thiol-cocatalyzed oxidant-free HAT hydrogenation of alkenes[37]. Nonetheless, stoichiometric silane is required as a reductant in their catalytic system.

We were motivated to devise silane- and peroxide-free catalytic HAT hydrogenation reactions in order to expand the scope of HAT hydrogenations beyond the canonical synthesis of complex molecules. We hypothesized that this might be possible by using ascorbic acid[38–41] and a combined cobalt/photoredox catalysis[42–52] (Fig. 1d). Inspired by König's proposed mechanism for the cobalt/photoredox-catalyzed alkene isomerization[48], we designed our HAT hydrogenation by metal-photoredox cooperative catalysis[53–57] as shown in Fig. 2a. The photoredox catalyst[58–62] (PC) is excited by visible light and oxidizes ascorbate[38], which exists in equilibrium with ascorbic acid, to afford the reduced PC and the corresponding ascorbic acid radical. The reduction of cobalt(II) with the reduced PC generates cobalt(I), which is converted to cobalt(III) hydride upon protonation. HAT from the cobalt(III) hydride to an unactivated alkene should regenerate the cobalt(II) catalyst and afford an alkyl radical. The second HAT to the resulting alkyl radical from ascorbic acid should realize the hydrogenation of the alkene. The resulting ascorbic acid radicals disproportionate to ascorbic acid and dehydroascorbic acid (Fig. 2b)[63].

## Results

**Investigation of reaction conditions**. Based on the proposed reaction design, we started to examine suitable cobalt-based co-catalysts for the photoredox-mediated HAT hydrogenation. Initially, we attempted the hydrogenation of **1a** using Ru(bpy)₃Cl₂ as the PC and ascorbic acid as a hydrogen source. Hydrogenated **2a** was not observed in the presence of cobalt chloride (Table 1, entry 1) or Co(acac)₂·2H₂O (acac: 2,4-pentanedionato), which is

**Table 1 Evaluation of the reaction conditions for the combined cobalt/photoredox-catalyzed hydrogenation of 1a.**

| Entry | Co complex (X mol%) | Y (mol%) | Yield (%)[a] |
|---|---|---|---|
| 1 | CoCl$_2$·6H$_2$O (5.0) | 1.0 | 0 |
| 2 | Co(acac)$_2$·2H$_2$O (5.0) | 1.0 | 0 |
| 3 | **3a** (5.0) | 1.0 | 13 |
| 4 | **3b** (5.0) | 1.0 | 6 |
| 5 | **3c** (5.0) | 1.0 | < 5 |
| 6 | **3d** (5.0) | 1.0 | 28 |
| 7 | **3d** (10) | 2.0 | 36 |
| 8 | **3e** (10) | 2.0 | 47 |
| 9 | **3f** (10) | 2.0 | 62 |
| 10[b,c] | **3f** (10) | 2.0 | 87 |
| 11[b,c,d] | **3f** (10) | 2.0 | 90 (92)[e] |
| 12[b,c,d] | none | 2.0 | 0 |
| 13[b,c,d] | **3f** (10) | 0 | 0 |
| 14[b,c,d,f] | **3f** (10) | 2.0 | 0 |
| 15[b,c,d,g] | **3f** (10) | 2.0 | 0 |

bpy = 2,2′-bipyridyl; acac = 2,4-pentanedionato. Unless otherwise noted, all reactions were carried out as follows: **1a** (0.10 mmol), Co complex (X mol%), Ru(bpy)$_3$Cl$_2$·6H$_2$O (Y mol%) and ascorbic acid (3.0 equiv); in 2-propanol/H$_2$O (3:1, 0.05 M); room temperature; 18 h; under blue LED irradiation (one panel); under Ar.
[a]Determined by [1]H NMR of the crude reaction mixture.
[b]PCy$_3$ (20 mol%) was added.
[c]In 2-propanol/H$_2$O (3:1, 0.2 M).
[d]With **1a** (0.20 mmol) under blue LED irradiation (two panels) and temperature control (ca. 25 °C).
[e]Isolated yield.
[f]In the dark.
[g]Without ascorbic acid.

the optimal catalyst for the cobalt/photoredox-catalyzed isomerization of alkenes[48] (entry 2). However, **2a** was obtained when salcomine **3a** (entry 3) or cobalt porphyrin **3b** (entry 4) was used as a cocatalyst. A further evaluation of cobalt salen complexes revealed that a complex with cyclohexanediamine linker (**3c**) showed a very low catalytic performance (entry 5), while a complex with phenylenediamine linker (**3d**) generated **2a** in higher yield (entry 6). The hydrogenation was further facilitated by increasing the amount of the cobalt catalyst and the photocatalyst (entry 7). Subsequently, we investigated the effects of the substituents, and discovered that complexes that contain *tert*-butyl groups (**3e**) show high catalytic performance (entry 8). Eventually, **3f**, which was previously studied as a catalyst for electrocatalytic hydrogen evolution[64], was identified as an optimal cobalt cocatalyst (entry 9). Using **3f**, hydrogenated product **2a** was obtained in 87% yield when the reaction was carried out with 0.2 M of **1a** in the presence of catalytic amounts of tricyclohexylphosphine (entry 10). It is expected that the sterically

demanding phosphine additive facilitated the reaction by preventing the accumulation of catalytically inactive cobalt-alkyl species[31,65]. Finally, **2a** was obtained in 92% isolated yield on a larger reaction scale by increasing the amount of photon source (entry 11).

The involvement of each reaction component was confirmed by control experiments (entries 12–15). In the absence of Co complex (entry 12), photocatalyst (entry 13), light (entry 14) or ascorbic acid (entry 15), no hydrogenation proceeded, supporting the proposed reaction design described in Fig. 2.

**Substrate scope**. The scope of the HAT hydrogenation of alkenes using the combined cobalt/photoredox catalysis is summarized in Table 2. Substrates with electron-rich arenes substituted with methyl (**2b**) and methoxy group (**2c**) were hydrogenated in high yield. Electron-deficient arenes containing trifluoromethyl and cyano groups were not affected during the hydrogenation and the hydrogenated products (**2d**, **2e**) were obtained in 86% and 87%

## Table 2 Scope of the ascorbic-acid-mediated hydrogenation of alkenes.

Reaction scheme: **3f** (10 mol%), PCy₃ (20 mol%), Ru(bpy)₃Cl₂ (2.0 mol%), ascorbic acid (3.0 equiv), 2-propanol/H₂O, 25 °C, Blue LED, 9-48 h. Substrate **1** → product **2**.

**2a**, 92%  **2b**, 92%  **2c**, 91%  **2d**, 86%

**2e**, 87%  **2f**, 76%[a,b]  **2g**, 78%  **2h**, 71%

**2i**, 67%[b]  **2j**, 78%[a,b]  **2k**, 87%  **2l**, 88%

**2m**, 80%  **2n**, 85%  **2o**, 87%  **2p**, 86%[b]

**2q**, 64%  **2r**, 45%[c]  **2s**, 65%[c]  **2t**, 79%  **2u**, 84%[e]

**2v**, 84%[d]  From Gln **2w**, 84%[e]  From Ser **2x**, 89%[d]  From Met **2y**, 85%[e]

From Gly-Gly **2z**, 79%  (−)-Isopregol **2aa**, 78%  Dihydrolinalool **2ab**, 93%  (−)-β-Citronellol **2ac**, 80%[d]

Mycophenolic Acid **2ad**, 86%[e,f]  Capsaicin **2ae**, 95%[e,f]

**2af**, 90%  **2ag**, 92%  **2ah** (IBCG), 89%

Cy cyclohexyl, bpy 2,2′-bipyridyl, Ac acetyl, Bn benzyl, Ts 4-toluenesulfonyl, Cbz benzyloxycarbonyl, TBDPS *tert*-butyldiphenylsilyl.
Isolated yields.
[a]2-Propanol/DMF/H₂O=3:3:2 was used as a solvent.
[b]40 °C.
[c]The yield and diastereoselectivity (*trans/cis*=75:25 for **2r**) were determined by GCMS analysis.
[d]The yield was determined by ¹H NMR analysis of the purified material containing **2** and **1**.
[e]Hydrogenation was performed for two cycles.
[f]Isolated after treatment with trimethylsilyldiazomethane.

**Table 3 Comparison of the HAT hydrogenation performance for the preparation of IBCG 2ah.**

| Entry | Conditions (mol%) | Yield (%)[a] |
|---|---|---|
| 1 | **3f** (10), $PCy_3$ (20), $Ru(bpy)_3Cl_2$ (2.0), ascorbic acid (300), 2-propanol/$H_2O$, 25 °C, Blue LED | 88 (89)[b] |
| 2[c] | $Mn(dpm)_3$ (10), $PhSiH_3$ (100), $t$BuOOH (150), 2-propanol, 22 °C | 35 |
| 3[d] | $Co(acac)_2$ (25), $PCy_3$ (25), $Et_3SiH$ (500), $t$BuOOH (25), DTBMP (50), 1,4-CHD (500), 1-propanol, 50 °C. | 5 |
| 4 | Pd/C (1.6), $H_2$ (1 atm), EtOH/$H_2O$, 25 °C | 96 |

For experimental details, see the Supplementary Information.
Cy cyclohexyl, bpy 2,2′-bipyridyl, dpm 2,2,6,6-tetramethyl-3,5-heptanedionato, acac 2,4-pentanedionato, DTBMP 2,6-di-*tert*-butyl-4-methylpyridine, 1,4-CHD 1,4-cyclohexadiene.
[a]Determined by [1]H NMR analysis of the crude reaction mixture.
[b]Isolated yield.
[c]Ref. 29.
[d]Ref. 31.

yield, respectively. Despite the putative intermediacy of a low-valent cobalt catalyst, $C(sp^2)$–Cl, $C(sp^2)$–Br and $C(sp^2)$–I bonds of haloarenes were compatible with the applied reaction conditions and the chemoselective hydrogenation of alkenes proceeded in good yield (**2f**, **2g**, **2h**). These results might be due to the fast protonation of the low-valent cobalt by ascorbic acid compared to the abstraction of halogen atoms. A substrate with an aromatic ketone was converted into the corresponding hydrogenated product in 67% yield (**2i**). Benzyl groups attached to oxygen or nitrogen remained intact and the desired hydrogenated products were obtained in 78% (**2j**) and in 87% yield (**2k**), again demonstrating a distinct chemoselectivity between the conditions used in this study and conventional heterogeneous palladium-catalyzed hydrogenation conditions. An aliphatic ketone in the substrate remained intact during the hydrogenation to afford **2l** in 88% yield. Substrates containing sulfone amide or benzyl carbonate moieties were also hydrogenated in good yield (**2m**, **2n**). **2o** was obtained in 87% yield while its $C(sp^3)$–Cl bond remained untouched, again suggesting that HAT hydrogenation outcompeted halide abstraction. In addition to trisubstituted alkenes, 1,1-disubstituted alkene derived from γ,δ-unsaturated alcohol was efficiently hydrogenated in 86% yield (**2p**). The hydrogenation of α,β-unsaturated esters also proceeded in 64% yield under the standard conditions (**2q**). The hydrogenation of an exocyclic alkene (**1r**) afforded the *trans* diastereomer of **2r** as the major product, which is consistent with the thermo-dynamically controlled stereoselectivity of the HAT hydrogenation. The aqueous solvent system required for this catalytic hydrogenation should be partly responsible for low yield of **2r**. Nevertheless, hydrogenation of alkenes without polar functional groups proceeded in synthetically useful yield (**2s**, **2t**, **2u**). γ,δ-Unsaturated amide was hydrogenated in 84% yield (**2v**). High functional-group tolerance of the present HAT hydrogenation was further demonstrated by the hydrogenation of amino-acid derivatives. A glutamine derivative was hydrogenated in 84% yield, indicating that the hydrogenation was not affected in the presence of a primary amide moiety (**2w**). The free hydroxyl group of serine and the thioether group of methionine were also compatible with this hydrogenation and afforded the desired products in 89% (**2x**) and 85% yield (**2y**). The substrate derived from glycylglycine was hydrogenated in 79% yield (**2z**), indicating applicability of the present hydrogenation to dipeptides. Given that trisubstituted alkenes are frequently found in terpenes, we also studied the hydrogenation of these natural products. We found that (−)-isopregol, dihydrolinalool, and (−)-β-citronellol were hydrogenated to afford **2aa**-**2ac** in 78–93% yield using the standard protocol. Due to the high compatibility of the present

HAT hydrogenation with a variety of functional groups, we next examined the potential late-stage hydrogenation of bioactive small molecules. An immunosuppressant mycophenolic acid with free carboxyl and phenol groups was hydrogenated to furnish **2ad** in 86% yield. It is noteworthy that the 1,2-disubstituted alkene of an analgesic agent capsaicin was hydrogenated to afford **2ae** in 95% yield. The catalytic hydrogenation was readily applicable to C-glycosides derived from ribofuranose (**2af**) and galactose (**2ag**). Importantly, the present protocol is even suitable for the hydrogenation of an unprotected galactose derivative and afforded the gene inducer isobutyl-β-C-galactoside (IBCG, **2ah**)[66] in 89% yield.

**Comparison with previous HAT hydrogenation protocols**. Despite the abundance of sugar derivatives in biologically active substances, HAT hydrogenations that were performed in the presence of unprotected sugar derivatives are, to the best of our knowledge, not known. Such protecting-group-free transformation[67] has deemed a challenge partly due to the chelating nature of the sugar derivatives, which can poison metal-based HAT catalysts. We speculated that **3f**, which does not possess vacant *cis*-coordination sites because of its planar tetradentate salophen ligand, might serve as a good metal-based catalyst for the HAT transformation of unprotected sugar derivatives. Indeed, the present system exhibits superior performance for the HAT hydrogenation of **1ah** relative to previously reported catalytic methods (Table 3, entries 2 and 3). Both the manganese catalyst with bidentate dipivaloylmethane ligands[29] and the cobalt catalyst with bidentate acetylacetone ligands[31] afford **2ah** in lower yield compared to the HAT hydrogenation catalyzed by **3f**. For a fair comparison, catalytic hydrogenation of **1ah** can be performed in 96% yield by heterogeneous palladium-catalyzed hydrogenation (entry 4). Nevertheless, the dual cobalt and photoredox-catalyzed HAT hydrogenation potentially serve as a chemoselective alternative for hydrogenation of alkenes in carbohydrate chemistry.

**Electrochemical study of the cobalt catalyst**. Electrochemical analysis of **3f** was conducted in order to assess our mechanistic proposal. Cyclic voltammetry of **3f** in DMF afforded quasi-reversible voltammogram (Fig. 3a) with $E_{1/2} = -1.36$ V vs. SCE, which reasonably corresponds to the $Co^{II}/Co^I$ redox couple[64]. The $E_{1/2}$ value suggests that the electron transfer to **3f** from the reduced photocatalyst ($E_{red}(Ru^{II}/Ru^I) = -1.33$ V vs. SCE)[68] is within an accessible range.

An increase in the voltammetric current in the voltammogram of **3f** was observed when increasing amounts of ascorbic acid

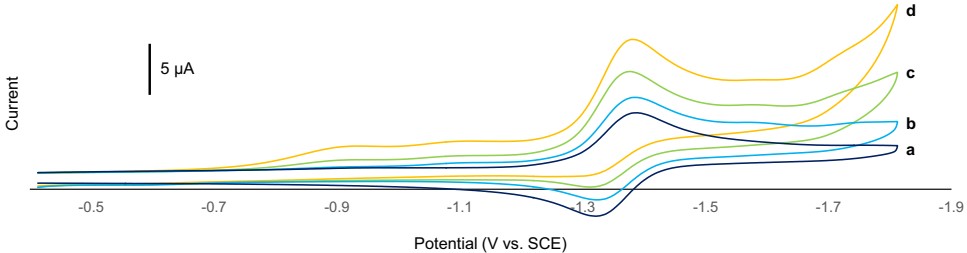

**Fig. 3 Cyclic voltammograms of 3f in the presence of various amounts of ascorbic acid.** Recorded at 100 mV/s in a DMF solution of **3f** (0.50 mM) and Bu$_4$NClO$_4$ (0.1 M). **a** In the absence of ascorbic acid. **b** 0.63 mM of ascorbic acid. **c** 1.5 mM of ascorbic acid. **d** 3.0 mM of ascorbic acid. The potential was corrected using Fc/Fc$^+$ as an internal standard; SCE saturated calomel electrode, Fc ferrocene.

**Fig. 4 Evaluation of the proposed mechanism by electrochemical analysis. a** Interpretation of the observations in cyclic voltammetry. **b** Plausible mechanisms of the cobalt-photoredox catalyzed hydrogenation in analogy with the electrochemical reaction; SCE saturated calomel electrode, AscH$_2$ ascorbic acid.

were added (Fig. 3b–d), while cathodic peak potentials remained similar as the ratio of ascorbic acid/**3f** increased. These changes in the voltammograms were in accord with reported electrochemical behavior of **3f** in the presence of increasing amounts of acetic acid[64] and indicate the generation of cobalt hydride intermediate from **3f** via cathodic reduction followed by protonation by ascorbic acid.

Thus, the electrochemical studies support the two processes described in Fig. 4a: (i) One electron reduction of **3f** by an electron transfer from the electrode, and (ii) Successive protonation of the reduced cobalt catalyst by ascorbic acid. In the absence of an alkene, the resulting cobalt hydride should be consumed by proton reduction[64]. In turn, in the photocatalytic reaction process, it is likely that the cobalt hydride, generated via one-electron transfer from the reduced ruthenium photocatalyst followed by protonation by ascorbic acid, reacts with an alkene to afford the carbon radical intermediate (Fig. 4b).

**Theoretical and experimental evaluation of the HAT pathway.** In order to gain insight into the HAT process, quantum chemical calculations were performed using the cobalt(salophen) hydride complex derived from **3f** and isobutene as a model substrate. The calculated pathway using UωB97X-D density functional [PCM (2-Propanol), LANL2DZ(f) for Co, 6-31G(d,p)] (see Section 5 of Supplementary Information for details) revealed that the barrier for the exergonic HAT between the two starting materials (SM) is only 7.3 kcal/mol higher in free energy relative to the SM at 25 °C (Fig. 5a), supporting that the HAT should proceed rapidly under the current reaction conditions. In the computed

transition state, the cobalt center possesses square pyramidal geometry without *cis*-vacant sites (Fig. 5b). This molecular geometry closely resembles the reported transition states of the HAT between alkenes and cobalt(porphyrin) hydride complexes[69,70]. The observed similarity suggests that, like the [H-Co$^{III}$(porphyrin)] complex, the [H-Co$^{III}$(salophen)] complex favors HAT compared to concerted migratory insertion when reacting with alkenes.

As for an experimental support for HAT, the intermediacy of an alkyl radical in the hydrogenation reaction was confirmed by a radical-trapping experiment. When the hydrogenation of **1ac** was attempted in the presence of TEMPO, no hydrogenated product **2ac** was obtained. Instead, TEMPO adduct **4** was unambiguously observed by LCMS analysis of the reaction mixture (Fig. 6, see Sections 2–5 of Supplementary Information for details). This observation is in accord with the intermediacy of tertiary alkyl radical presumably generated by HAT from cobalt hydride to **1ac**.

**Discussion**
In summary, we have developed a combined cobalt/PC system that enables the silane- and peroxide-free, ascorbic-acid-mediated HAT hydrogenation of alkenes in aqueous media. While the aqueous conditions were generally beneficial for the hydrogenation of polar substrates, substrates without polar functional groups were also hydrogenated in synthetically useful yield. The present reaction offers not only a more sustainable and safer alternative to previously reported HAT hydrogenation methods, but also a highly functional-group tolerant hydrogenation protocol suitable for the late-stage hydrogenation of amino-acid

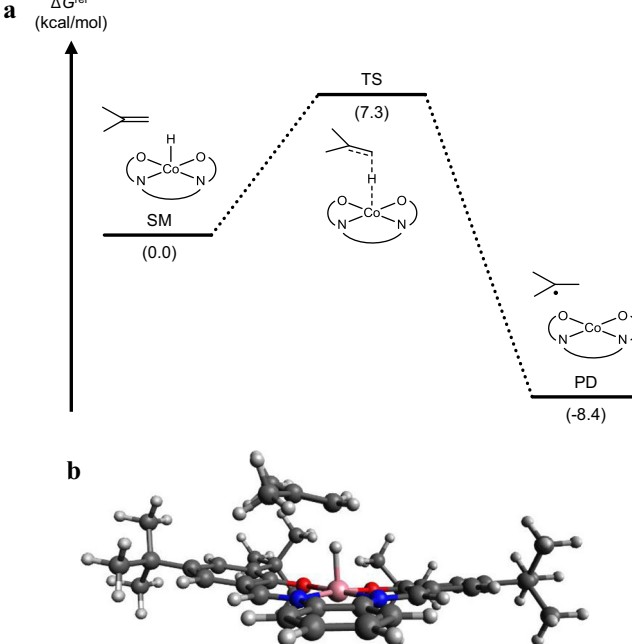

**Fig. 5 HAT pathway elucidated by quantum chemical calculations.**
**a** Computed energy diagram of the HAT between cobalt(salophen) hydride complex and isobutene. **b** Computed structure of the transition state of the HAT between cobalt(salophen) hydride complex and isobutene; $\Delta G^{rel}$ relative free energy, SM starting materials, TS transition state, PD products. See the Supplementary Information for computational details.

**Fig. 6 Detection of the radical intermediate by TEMPO trapping.** The formation of TEMPO adduct supports the presence of alkyl radical intermediate which is likely to be formed by HAT to **1ac**; Cy cyclohexyl, bpy 2,2′-bipyridyl, TEMPO (2,2,6,6-Tetramethylpiperidin-1-yl)oxyl.

derivatives, terpenes, and drug molecules. The direct hydrogenation of an unprotected galactose derivative should be noted with regard to the potential future prospects for the HAT functionalization of unprotected sugar derivatives. The feasibility of the proposed HAT process was supported by electrochemical analysis, theoretical investigation and the radical-trapping experiment. Applications of this catalytic system to other Markovnikov-selective hydrofunctionalization reactions of alkenes are currently in progress in our group.

## Methods
**General procedure for ascorbic-acid-mediated HAT hydrogenation of alkenes.**
In an argon-filled glove box, a flame dried reaction vial was charged with an alkene **1** (0.20 mmol), ascorbic acid (106 mg, 0.60 mmol), **3f** (12.0 mg, 20 μmol), tricyclohexylphosphine (11.2 mg, 40 μmol) and Ru(bpy)$_3$Cl$_2$•6H$_2$O (3.0 mg, 4.0 μmol).

The vial was capped and removed from the glove box. A mixed solvent (2-propanol/H$_2$O = 3:1, 1 mL) was added to the vial via syringe, and the syringe hole was carefully sealed with a vinyl tape. The reaction vial was placed in front of the light source (ca. 3 cm from two blue LED panels) in a cold room (4 °C) so that the temperature of the reaction mixture was kept approximately at 25 °C. After stirring for the indicated time, the reaction mixture was cooled in an ice bath and sat. aq. NaHCO$_3$ was added. Organic material was extracted with EtOAc (x 3) and the combined organic layer was washed with brine. The organic layer was concentrated under reduced pressure and purified by column chromatography (silica gel) to afford the hydrogenated product **2**.

## Data availability
The authors declare that all other data supporting the findings of this study are available within the article and Supplementary Information files, and also are available from the corresponding author upon reasonable request.

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

## Acknowledgements

This work was supported in part by JSPS KAKENHI Grant Number JP17H03049 and JP20H02730 (to S. Matsunaga), JSPS KAKENHI Grant Number JP19K21218 and JP20K15946 (to M.K.), JSPS-WPI and JST-ERATO (No. JPMJER1903) (to S. Maeda), as well as Platform Project for Supporting Drug Discovery and Life Science Research (BINDS) from AMED under Grant No. JP20am0101093 (to common HRMS facilities). M.K. learned and performed quantum chemical calculations at the Institute for Chemical Reaction Design and Discovery (ICReDD), Hokkaido University, which was established by World Premier International Research Initiative (WPI), MEXT, Japan. We thank Dr. Kenichi Matsuda and Prof. Dr. Toshiyuki Wakimoto in Hokkaido University for allowing access to their lyophilization machine. We thank Dr. Hideo Takakura and Prof. Dr. Mikako Ogawa in Hokkaido University for their support in the Stern-Volmer quenching studies. We also thank Dr. Takuro Suzuki in our group for initially providing **3d**. Y.K. thanks the Kawamura Scholarship Foundation for a fellowship.

## Author contributions

Y.K., Y.S., Y.Y., and M.K. performed the experiments and analyzed the data. S. Maeda and M.K. performed the quantum chemical calculations. Y.K., T.Y., S. Maeda, M.K., and S. Matsunaga conceived and designed the experiments and prepared the manuscript. All authors contributed to discussions and commented on the manuscript.

## Competing interests

The authors declare no competing interests.
