## [Peer Review File · Nature Communications]

REVIEWER COMMENTS

Reviewer #1 (Remarks to the Author):

In this submission the authors described catalytic hydrogenation of olefins via CoH-mediated HAT under photoredox conditions. The merge between the photoredox cycle and the CoH HAT cycle is realized by the reduction of Co(II) to Co(I) by Ru(I). This approach uses solely ascorbic acid as the reducing agent, which renders the reported system substantially greener than previously disclosed silane-based HAT hydrogenation systems. Applications on sugar derivatives is demonstrated and compared to the reported Co/Mn-based systems. In addition, the authors provide electrochemical evidence to support the proposed key ET/PT process. There are some concerns (see below) on the mechanistic aspects, nonetheless overall this is a decent contribution to the CoH HAT chemistry with potential in various synthetic applications. I would like to recommend publication given the authors could kindly address the following questions/comments:

- (1) It is suggested that control experiments leaving out either (a) blue light exposure, or (b) Ru should be included in table 1 (CoSalen absorbs some blue light too).
- (2) Although readily available Ru(bpy)₃Cl₂ serves as a good catalyst, it is suggested the authors test a couple of alternative photocatalysts that are either more or less reducing for comparison, such that the conclusion on the ET can be strengthened.
- (3) Isopropanol is employed as the optimal solvent in previous reports (e.g. Shenvi group) partly because of it activates the silane. However, the use of IPA in silane-free reductions raises questions on the potential involvement of IPA as the real reducing agent. It is suggested that either D-labeling, reactions in alternative solvent, or other experiments should be performed to investigate this possibility.
- (4) As suggested the proposed catalytic cycle, other phenol-like reducing agents (e.g. BHT) should also be viable. It would be good to show one of two addition examples.
- (5) It is suggested to include an entry of classic precious metal catalyzed hydrogenation reaction in table 4 for fair comparison, as such hydrogenations are widely used in carbohydrate synthesis and in principle not easily inhibited by the substrate. It would also be nice to see if this section could be expanded a bit to include a few more examples of carbohydrate derivatives.
- (6) It is noted a relevant reference involving HAT reactions catalyzed by CoSalen and Ru(bpy)₃Cl₂ seems missing: Sun et al. ACS Catal. 2020, 10, 4983.

Reviewer #2 (Remarks to the Author):

The present ms by Kojima, Matsunaga and co-workers describes a dual photoredox/cobalt approach for the hydrogenation of alkenes. The work is interesting and the investigated topic of high novelty, deserving to be considered for publication in the present journal. However, the claims offered by the Authors in the ms are not fully supported by the experimental data, particularly under the mechanistic standpoint.

Accordingly, I am convinced that this ms may be accepted for publication in Nat. Commun. only if the Authors fully demonstrate the mechanism involved in the present contribution. Specific issues to be considered upon revision are reported below in decreasing order of importance.

- The key step of the proposed mechanism is a HAT from Co(III)-H to the alkene reported in Figure 2. Unfortunately, no direct evidence of the occurrence of this step is offered by the Authors. Also literature precedents (see Ref. 48 by Koenig's group) offer a different mechanistic pathway, with addition of a Co(III)-H species across a C=C double bond. The Authors must provide striking evidence demonstrating the occurrence of a HAT step (e.g. by trapping of a relevant intermediate and/or its spectroscopic detection; computational data may also help here). Along the same line, it is important to consider different scenarios, as those reported in: Chem. Sci. 2018,9, 4977, since H₂ formation may occur also in the present system (see further below).

- Table 1. I think that the optimization of reaction conditions needs to be significantly improved. Did the Authors perform suitable blank experiments (reaction performed in the dark / in the absence of the photocatalyst / in the absence of the Co-based catalyst / in the absence of ascorbic

acid)? Which is the role of the adopted phosphine ligand? This is never mentioned in the text!

- Table 1. The chosen reaction medium is another point that would need more attention. In particular, the Authors did not perform any screening and used exclusively a iPrOH/H₂O mixture. Did the Authors try to use a non-protic solvent? Is it possible to observe the formation of H₂ gas (via protonation of in situ formed Co(III)-H species)? This would open the path to a different mechanistic interpretations.

- Table 2. It appears that the substrates need to be substituted with a polar handle (often a N-H moiety or at least a carbonyl/carboxyl group). Is this required to observe the desired reactivity? Did the Authors try to use apolar substrates (e.g. unsubstituted olefinic hydrocarbons)?

- Mechanistic studies. A more convincing depiction of the reaction mechanism would require to evaluate the actual quenching, if any, of the photocatalyst excited state by the different reaction partners (e.g. ascorbic acid, the Co-based catalyst, the substrate, the employed additives, ...).

- Figure 3. I would recommend to supplement the reported study by using the actual reaction mixture (if feasible) or at least by adding a small portion of a protic solvent to see how the CV trace is affected under these conditions.

Reviewer #3 (Remarks to the Author):

Chemoselective hydrogen atom transfer (HAT) to unactivated alkenes using photoredox/Co catalysis has been reported in this manuscript. Conventional HAT methods are dependent upon stoichiometric use of silanes and peroxides which limits the practicality of such methodologies. This methodology uses ascorbic acid as stoichiometric transfer hydrogenation reagent. Cobalt-salen complex in combination with a Ru-photocatalyst is able to facilitate the HAT process in aqueous media. High functional group tolerance, application in late stage hydrogenation of amino acids, and drug intermediates as well as hydrogenation of unprotected sugar molecules are advantages of this synthetic methodology. A mechanism based on hydrogen atom transfer has been proposed based on CV studies. Overall, the manuscript is well written and has important results of broad interest and provides a solution to existing methods which have shortcomings.

I think there are several ways to improve the manuscript. Some suggestions and comments are provided below.

Comments:

There is no discussion of the results and what is presented under discussion is really conclusion (s).

Question/comment:

Most of the substrates used in the study have either hydrogen bond donors or acceptors. The authors should comment on the nature of the substrate used in the study.

The solvent in these studies is water/isopropanol. Do other water miscible solvents such as acetone effective?

It is interesting that the reaction is compatible with substrates with halogen atoms. Given the nature of Co to abstract halogen atoms, the author should comment on why their method is chemoselective. How about aliphatic compounds with a halogen atom? Are they compatible?

There is only one example of a hydrophobic substrate (2q). Are there more examples which work? I think this is where solvents can play a role.

Is C-functionalized acetylated glucose (or similar substrate) a competent substrate for the reaction?

The solvent combination is a potential limitation for substrates which are hydrophobic.

List of Revisions and Responses to Reviewers

[Responses to Reviewer #1]

In this submission the authors described catalytic hydrogenation of olefins via CoH-mediated HAT under photoredox conditions. The merge between the photoredox cycle and the CoH HAT cycle is realized by the reduction of Co(II) to Co(I) by Ru(I). This approach uses solely ascorbic acid as the reducing agent, which renders the reported system substantially greener than previously disclosed silane-based HAT hydrogenation systems. Applications on sugar derivatives is demonstrated and compared to the reported Co/Mn-based systems. In addition, the authors provide electrochemical evidence to support the proposed key ET/PT process. There are some concerns (see below) on the mechanistic aspects, nonetheless overall this is a decent contribution to the CoH HAT chemistry with potential in various synthetic applications. I would like to recommend publication given the authors could kindly address the following questions/comments:

Comment 1-1:

“(1) It is suggested that control experiments leaving out either (a) blue light exposure, or (b) Ru should be included in table 1 (CoSalen absorbs some blue light too).”

Our response:

Control experiments in the absence of blue LED irradiation or ruthenium-based photocatalyst were conducted, and it was confirmed that both were necessary for the catalytic hydrogenation. The results of the control experiments were described in entries 13 and 14 of the revised Table 1. These findings are also mentioned in the main text.

Table 1. Evaluation of the reaction conditions for the combined cobalt/photoredox-catalyzed hydrogenation of 1a ^a			
			
Entry	Co complex (X mol%)	Y	Yield (%) ^b
13 ^{c,d,e}	3f (10)	0	0
14 ^{c,d,e,g}	3f (10)	2.0	0

Part of revised Table 1:

Added text:

The involvement of each reaction component was confirmed by control experiments (entries 12-15). In the absence of Co complex (entry 12), photocatalyst (entry 13), light (entry 14) or ascorbic acid (entry 15), no hydrogenation proceeded, supporting the proposed reaction design described in Figure 2.

Comment 1-2:

“(2) Although readily available Ru(bpy)₃Cl₂ serves as a good catalyst, it is suggested the authors test a couple of alternative photocatalysts that are either more or less reducing for comparison, such that the conclusion on the ET can be strengthened.”

Our response:

The relationship between the choice of photocatalyst and the efficiency of the hydrogenation was studied, and we confirmed that more reducing photocatalyst certainly promoted the hydrogenation in higher efficiency. These results were summarized in Table S1 and explained in section 3-1 of Supplementary Information.

Added Table S1:

Table S1. Effect of other photocatalysts in the hydrogenation of **1a**

Entry	Photocatalyst	NMR Yield of 2a (%)	Ered (PC/PC ⁻) [V vs SCE]
1	Ru(bpy) ₃ Cl ₂	90	-1.33
2	rhodamine 6G	0	-1.14
3	4CzIPN	58	-1.21
4	[Ir(dF(CF ₃)ppy) ₂ (dtbbpy)]PF ₆	38	-1.37
5	[Ir(ppy) ₂ (dtbbpy)]PF ₆	>99	-1.51

Comment 1-3:

“(3) Isopropanol is employed as the optimal solvent in previous reports (e.g. Shenvi group) partly because of it activates the silane. However, the use of IPA in silane-free reductions raises questions on the potential involvement of IPA as the real reducing agent. It is suggested that either D-labeling, reactions in alternative solvent, or other experiments should be performed to investigate this possibility.”

Our response:

Investigation of the hydrogenation in other solvents were performed, and it was confirmed that 2-propanol was not necessary to obtain the desired product in high yield. These results suggest that 2-propanol is unlikely to be a real reductant in this catalytic system. These results were summarized in Table S2 and explained in section 3-2 of Supplementary Information.

Added Table S2:

Table S2. Effect of other solvents in the hydrogenation of **1a**

Entry	Solvent	NMR Yield of 2a (%)
1	2-propanol/H ₂ O (3:1)	90
2	2-methyl-2-propanol/H ₂ O (3:1)	83
3	CH ₃ CN/H ₂ O (3:1)	85
4	DMF/H ₂ O (3:1)	70
5	acetone/H ₂ O (3:1)	79
6	2-propanol	0
7	CH ₃ CN	4
8	DMF	4

Comment 1-4:

“(4) As suggested the proposed catalytic cycle, other phenol-like reducing agents (e.g. BHT) should also be viable. It would be good to show one of two addition examples.”

Our response:

Catalytic hydrogenation using phenol derivatives instead of ascorbic acid was attempted and it was found that although gallic acid slightly promoted the hydrogenation, the reactivity was much lower than the reaction using ascorbic acid. These observations were summarized in Table S3 and explained in section 3-3 of the Supplementary Information.

Added Table S3:

Entry	Reductant	NMR Yield of 2a (%)
1	ascorbic acid	90
2	BHT	0
3	gallic acid	<5

Comment 1-5-1:

“(5) It is suggested to include an entry of classic precious metal catalyzed hydrogenation reaction in table 4 for fair comparison, as such hydrogenations are widely used in carbohydrate synthesis and in principle not easily inhibited by the substrate.”

Our response:

We experimentally confirmed that Pd/C-catalyzed hydrogenation afforded **2ah** in high yield. This result was added in entry 4 of the revised Table 3. Considering the distinct chemoselectivity of the ascorbic acid-mediated HAT hydrogenation, we nonetheless assume that the present protocol would be beneficial as a chemoselective alternative in the field of carbohydrate chemistry. We added discussion regarding such aspects in the main text.

Revised Table 3:

Table 3. Comparison of the HAT hydrogenation performance for the preparation of IBCG 2ah ^a		
Entry	Conditions (mol%)	Yield (%) ^b
1	3f (10), PCy ₃ (20), Ru(bpy) ₃ Cl ₂ (2.0), ascorbic acid (300) 2-propanol/H ₂ O, 25 °C, Blue LED	88 (89) ^c
2 ^d	Mn(dpm) ₃ (10), PhSiH ₃ (100), t BuOOH (150) 2-propanol, 22 °C	35
3 ^e	Co(acac) ₂ (25), PCy ₃ (25), Et ₃ SiH (500), t BuOOH (25), DTBMP (50), 1,4-CHD (500) 1-propanol, 50 °C	5
4	Pd/C (1.6), H₂ (1 atm) EtOH/H₂O, 25 °C	96

^aFor experimental details, see the Supplementary Information. ^bDetermined by ¹H NMR analysis of the crude reaction mixture. ^cIsolated yield. ^dRef. 29. ^eRef. 31; Cy = cyclohexyl; bpy = 2,2'-bipyridyl; dpm = 2,2,6,6-tetramethyl-3,5-heptanedionato; acac = 2,4-pentanedionato; DTBMP = 2,6-di-*tert*-butyl-4-methylpyridine; 1,4-CHD = 1,4-cyclohexadiene.

Added text:

For fair comparison, catalytic hydrogenation of **1ah** can be performed in 96% yield by heterogeneous palladium-catalyzed hydrogenation (entry 4). Nevertheless, the dual cobalt and photoredox-catalyzed HAT hydrogenation potentially serves as a chemoselective alternative for hydrogenation of alkenes in carbohydrate chemistry.

Comment 1-5-2:

“It would also be nice to see if this section could be expanded a bit to include a few more examples of carbohydrate derivatives.”

Our response:

We appreciate the insightful suggestion from Reviewer #1. As for a preliminary answer to this comment, hydrogenation of two additional carbohydrate derivatives (**2af**, **2ag**) were successfully performed and these results were added in revised Table 2. We are working on hydrogenation of more unprotected carbohydrate derivatives by our protocol and on comparison of reactivity with established methods, but we anticipate that these efforts could be beyond the scope of this revision. We continue working on the transformation of unprotected carbohydrates and would like to report the results in future in a following manuscript.

Added substrates in the revised Table 2:

Added text:

The catalytic hydrogenation was readily applicable to C-glycosides derived from ribofuranose (**2af**) and galactose (**2ag**).

Comment 1-6:

“(6) It is noted a relevant reference involving HAT reactions catalyzed by CoSalen and Ru(bpy)₃Cl₂ seems missing: Sun et al. ACS Catal. 2020, 10, 4983.”

Our response:

The report by Sun *et al.* was newly cited as reference 52.

Added reference:

52 Sun, H.-L., Yang, F., Ye, W.-T., Wang, J.-J. & Zhu, R. Dual cobalt and photoredox catalysis enabled intermolecular oxidative hydrofunctionalization. *ACS Catal.* **10**, 4983-4989 (2020).

[Responses to Reviewer #2]

The present ms by Kojima, Matsunaga and co-workers describes a dual photoredox/cobalt approach for the hydrogenation of alkenes. The work is interesting and the investigated topic of high novelty, deserving to be considered for publication in the present journal. However, the claims offered by the Authors in the ms are not fully supported by the experimental data, particularly under the mechanistic standpoint.

Accordingly, I am convinced that this ms may be accepted for publication in *Nat. Commun.* only if the Authors fully demonstrate the mechanism involved in the present contribution. Specific issues to be considered upon revision are reported below in decreasing order of importance.

Comment 2-1:

“- The key step of the proposed mechanism is a HAT from Co(III)-H to the alkene reported in Figure 2. Unfortunately, no direct evidence of the occurrence of this step is offered by the Authors. Also literature precedents (see Ref. 48 by Koenig's group) offer a different mechanistic pathway, with addition of a Co(III)-H species across a C=C double bond. The Authors must provide striking evidence demonstrating the occurrence of a HAT step (e.g. by trapping of a relevant intermediate and/or its spectroscopic detection; computational data may also help here). Along the same line, it is important to consider different scenarios, as those reported in: *Chem. Sci.* 2018,9, 4977, since H₂ formation may occur also in the present system (see further below).”

Our response:

As a response to this comment, a new section titled “Theoretical and experimental evaluation of the HAT pathway” was added to the main text.

<Theoretical investigation>

In order to elucidate the feasibility of the HAT process, quantum chemical calculation was conducted. As a result, transition state of HAT was found to be only 7.3 kcal/mol higher in free energy compared to starting materials. This low energy barrier suggests that HAT should readily proceed under the current reaction conditions. These results were summarized in newly added Figure 5 and explained in the main text.

Added Figure 5:

Figure 5. HAT pathway elucidated by quantum chemical calculations. a Computed energy diagram of the HAT between cobalt(salophen) hydride complex and isobutene. **b** Computed structure of the transition state of the HAT between cobalt(salophen) hydride complex and isobutene; SM = starting materials; TS = transition state; PD = products. See the Supplementary Information for computational details.

Added text:

In order to gain insight into the HAT process, quantum chemical calculations were performed using the cobalt(salophen) hydride complex derived from **3f** and isobutene as a model substrate. The calculated pathway revealed that the barrier for the exergonic HAT between the two starting materials (SM) is only 7.3 kcal/mol higher in free energy

relative to the SM at 25 °C (Figure 5a), supporting that the HAT should proceed rapidly under the current reaction conditions. In the computed transition state, the cobalt center possesses square pyramidal geometry without *cis*-vacant sites (Figure 5b). This molecular geometry closely resembles the reported transition states of the HAT between alkenes and cobalt(porphyrin) hydride complexes.^{69,70} The observed similarity suggests that, like the [H-Co^{III}(porphyrin)] complex, the [H-Co^{III}(salophen)] complex favors HAT compared to concerted migratory insertion when reacting with alkenes.

<Experimental investigation>

The presence of tertiary radical in the catalytic hydrogenation, which is a key intermediate in the HAT process, was experimentally confirmed by running the reaction in the presence of TEMPO. The formation of the TEMPO adduct **4** was confirmed by comparing its *m/z* and retention time in LCMS with those of **4** independently synthesized by a reported method. These studies were summarized in newly added Figure 6 and explained in the main text. The related experimental details were described in section 2-5 of Supplementary Information.

Added Figure 6:

Figure 6. Detection of the radical intermediate by TEMPO trapping.

Added text:

As for an experimental support for HAT, intermediacy of an alkyl radical in the hydrogenation reaction was confirmed by a radical trapping experiment. When the hydrogenation of **1ac** was attempted in the presence of TEMPO, no hydrogenated product **2ac** was obtained. Instead, TEMPO adduct **4** was unambiguously observed by LCMS analysis of the reaction mixture (Figure 6). This observation is in accord with the

intermediacy of tertiary alkyl radical presumably generated by HAT from cobalt hydride to **1ac**.

Comment 2-2-1:

“- Table 1. I think that the optimization of reaction conditions needs to be significantly improved. Did the Authors perform suitable blank experiments (reaction performed in the dark / in the absence of the photocatalyst / in the absence of the Co-based catalyst / in the absence of ascorbic acid)?”

Our response:

By additional blank experiments leaving out either blue LED, photocatalyst, Co catalyst or ascorbic acid, it was confirmed that all these components were indispensable for the catalytic hydrogenation. These results were summarized in entries 12-15 in the revised Table 1 and additionally explained in the main text. The title of the related section in the main text and the title of Table 1 were changed accordingly.

Part of revised Table 1:

Table 1. Evaluation of the **reaction conditions** for the combined cobalt/photoredox-catalyzed hydrogenation of **1a**^a

Entry	Co complex (X mol%)	Y	Yield (%) ^b
12 ^{c,d,e}	none	2.0	0
13 ^{c,d,e}	3f (10)	0	0
14 ^{c,d,e,g}	3f (10)	2.0	0
15 ^{c,d,e,h}	3f (10)	2.0	0

^aUnless otherwise noted, all reactions were carried out as follows: **1a** (0.10 mmol), Co complex (X mol%), Ru(bpy)₃Cl₂·6H₂O (Y mol%) and ascorbic acid (3.0 equiv.); in 2-propanol/H₂O (3:1, 0.05 M); room temperature; 18 h; under blue LED irradiation (one panel); under Ar. ^bDetermined by ¹H NMR of the crude reaction mixture. ^cPCy₃ (20 mol%) was added. ^dIn 2-propanol/H₂O (3:1, 0.2 M). ^eWith **1a** (0.20 mmol) under blue LED irradiation (two panels) and temperature control (ca. 25 °C). ^fIsolated yield. ^gIn the dark. ^hWithout ascorbic acid; bpy = 2,2'-bipyridyl; acac = 2,4-pentanedionato.

Added text:

The indispensable role of each reaction component was confirmed by control experiments (entries 12-15). In the absence of Co complex (entry 12), photocatalyst (entry 13), light (entry 14) or ascorbic acid (entry 15), no hydrogenation proceeded, supporting the proposed reaction design described in Figure 2.

Comment 2-2-2:

“Which is the role of the adopted phosphine ligand? This is never mentioned in the text!”

Our response:

According to the reports by Halpern and Herzon, the phosphine additive is expected to promote the homolysis of a cobalt-alkyl bond, thereby prevents the formation of

catalytically inactive alkylcobalt species. Such discussion was added in the main text and the report by Halpern was newly cited as reference 65.

Added text:

It is expected that the sterically demanding phosphine additive facilitated the reaction by preventing the accumulation of catalytically inactive cobalt-alkyl species.^{31,65}

Added reference:

65 Ng, F. T. T., Rempel, G. L. & Halpern, J. Steric influences on cobalt-alkyl bond dissociation energies. *Inorg. Chim. Acta* **77**, L165-L166 (1983).

Comment 2-3-1:

“- Table 1. The chosen reaction medium is another point that would need more attention. In particular, the Authors did not perform any screening and used exclusively a iPrOH/H₂O mixture. Did the Authors try to use a non-protic solvent?”

Our response:

Investigation of the hydrogenation in other solvents were performed, and it was confirmed that the hydrogenation in a non-protic solvent (CH₃CN or DMF) was sluggish. We hypothesized that the beneficial effect of aqueous solvent might derive from enhanced solubility and electrolytic dissociation of ascorbic acid in the reaction medium, but more studies are needed to verify the role of aqueous solvent. The results of reactions in non-protic solvents were described in entries 7 and 8 in Table S2 and discussed in section 3-2 of the Supplementary Information.

Part of added Table S2:

Table S2. Effect of other solvents in the hydrogenation of 1a		
		
Entry	Solvent	NMR Yield of 2a (%)
7	CH ₃ CN	4
8	DMF	4

Comment 2-3-2:

“Is it possible to observe the formation of H₂ gas (via protonation of in situ formed Co(III)-H species)? This would open the path to a different mechanistic interpretations.”

Our response:

Formation of H₂ gas from ascorbic acid by the dual cobalt and photoredox catalysis was observed by the parallel reactions using COWare. However, attempt of hydrogenation using the cobalt-photoredox catalysis under H₂ atmosphere did not afford the hydrogenated product. These results imply that H₂ gas might be generated but is not a major reductant in the current catalytic system.

The result of H₂ gas detection was summarized in Figure S3a and S3b and explained in section 4-3 of Supplementary Information. The attempt of H₂-mediated hydrogenation was summarized in Figure S4 and explained in section 4-4 of Supplementary Information.

Added Figure S3a and S3b:

Figure S3a. Detection of H₂ gas by the parallel reactions using COWare

Figure S3b. The setup of the reaction using COware

Added Figure S4:

Figure S4. Attempt of hydrogenation under H₂ atmosphere

Comment 2-4:

“- Table 2. It appears that the substrates need to be substituted with a polar handle (often a N-H moiety or at least a carbonyl/carboxyl group). Is this required to observe the desired reactivity? Did the Authors try to use apolar substrates (e.g. unsubstituted olefinic hydrocarbons)?”

Our response:

Additional investigation of substrates without polar functional groups were performed and the hydrogenation also proceed in synthetically useful yield for these substrates. We anticipate that, considering these experimental results and the theoretical studies, the present reaction is not likely to be facilitated by coordination of a polar functional group

to cobalt. The results of hydrogenation of nonpolar substrates (**2s**, **2t**, **2u**) were added to the revised Table 2 and these results were additionally discussed in the main text.

Added substrates in revised Table 2:

Added text:

Nevertheless, hydrogenation of alkenes without polar functional groups proceeded in synthetically useful yield (**2s**, **2t**, **2u**).

Comment 2-5:

“- Mechanistic studies. A more convincing depiction of the reaction mechanism would require to evaluate the actual quenching, if any, of the photocatalyst excited state by the different reaction partners (e.g. ascorbic acid, the Co-based catalyst, the substrate, the employed additives, ...).”

Our response:

Stern-Volmer quenching analysis of $\text{Ru}(\text{bpy})_3\text{Cl}_2$ was conducted using varying concentration of the reaction components. As a result, ascorbic acid was confirmed to quench the excited photocatalyst while the representative alkene substrate **1a** or tricyclohexylphosphine did not. The quenching study of Co-based catalyst was not able to be performed due to its strong absorption in visible light region. These results were summarized in Figure S1a and explained in section 4-1 of Supplementary Information.

Added Figure S1a:

Figure S1a. Results of the Stern-Volmer luminescence quenching analysis

Figure S1b. Absorption of **3f** (50 μ M in 2-propanol) in visible light region

Comment 2-6:

“- Figure 3. I would recommend to supplement the reported study by using the actual reaction mixture (if feasible) or at least by adding a small portion of a protic solvent to see how the CV trace is affected under these conditions.”

Our response:

Systematic analysis of cyclic voltammograms of **3f** in the mixed solvent (2-propanol/H₂O) was not successful due to the background current presumably derived from the solvent. Addition of a small amount of 2-propanol and H₂O to a solution of **3f**

in DMF did not make an obvious change in the cyclic voltammogram. This suggests that ascorbic acid might work as a better proton source for Co^{I} , but we cannot clearly determine the actual proton source in the reaction only by this electrochemical study considering the difference in stoichiometry between ascorbic acid and the protic solvent. The cyclic voltammogram in the presence of added protic solvent was shown in Figure S2 of Supplementary Information. The relevant discussion was added in section 4-2 of Supplementary Information.

Added Figure S2 in Supplementary Information:

Figure S2. Effect of a protic solvent for cyclic voltammogram of **3f**. Recorded at 100 mV/s in a DMF solution of **3f** (0.50 mM) and Bu_4NClO_4 (0.1 M). (a) In the absence of a protic solvent. (b) In the presence of 2-propanol (1.0 mM) and H_2O (0.33 mM). The potential was corrected using Fc/Fc^+ as an internal standard.

[Responses to Reviewer #3]

Chemoselective hydrogen atom transfer (HAT) to unactivated alkenes using photoredox/Co catalysis has been reported in this manuscript. Conventional HAT methods are dependent upon stoichiometric use of silanes and peroxides which limits the practicality of such methodologies. This methodology uses ascorbic acid as stoichiometric transfer hydrogenation reagent. Cobalt-salen complex in combination with a Ru-photocatalyst is able to facilitate the HAT process in aqueous media. High functional group tolerance, application in late stage hydrogenation of amino acids, and drug intermediates as well as hydrogenation of unprotected sugar molecules are advantages of this synthetic methodology. A mechanism based on hydrogen atom transfer has been proposed based on CV studies. Overall, the manuscript is well written and has important results of broad interest and provides a solution to existing methods which have shortcomings.

I think there are several ways to improve the manuscript. Some suggestions and comments are provided below.

Comment 3-1:

“There is no discussion of the results and what is presented under discussion is really conclusion (s).”

Our response:

The mentioned style of writing which present conclusions in the discussion section is widely appreciated in recent publications related to organic chemistry in *Nature Communications* (for example, *Nat. Commun.* **11**, Article number: 5480 (2020); *Nat. Commun.* **11**, Article number: 5500 (2020)). We have also prepared the manuscript following this way.

Comment 3-2:

“Most of the substrates used in the study have either hydrogen bond donors or acceptors. The authors should comment on the nature of the substrate used in the study.”

Our response:

A comment concerning the polarity of the substrates in this study was added in the discussion section of the main text.

Added text:

While the aqueous conditions were generally beneficial for hydrogenation of polar substrates, substrates without polar functional groups were also hydrogenated in synthetically useful yield.

Comment 3-3:

“The solvent in these studies is water/isopropanol. Do other water miscible solvents such as acetone effective?”

Our response:

In our additional investigation aqueous solvents containing *tert*-butanol, acetonitrile, DMF and acetone were effective for the desired hydrogenation. These findings were described in Table S2 and section 3-2 of Supplementary Information.

Part of added Table S2:

Entry	Solvent	NMR Yield of 2a (%)
1	2-propanol/H ₂ O (3:1)	90
2	2-methyl-2-propanol/H ₂ O (3:1)	83
3	CH ₃ CN/H ₂ O (3:1)	85
4	DMF/H ₂ O (3:1)	70
5	acetone/H ₂ O (3:1)	79

Table S2. Effect of other solvents in the hydrogenation of **1a**

Comment 3-4:

“It is interesting that the reaction is compatible with substrates with halogen atoms. Given the nature of Co to abstract halogen atoms, the author should comment on why their method is chemoselective. How about aliphatic compounds with a halogen atom? Are they compatible?”

Our response:

We anticipate that protonation of the low valent cobalt intermediate occurred faster than other side reactions. Additional experimentation revealed that the substrate containing a C(sp³)-Cl bond was also hydrogenated in 87% yield without abstraction of a chlorine atom. These discussions were added in the main text and the additional substrate **2o** was added in Table 2.

Added substrate in revised Table 2:**Added text (1):**

These results might be due to the fast protonation of the low-valent cobalt by ascorbic acid compared to abstraction of halogen atoms.

Added text (2):

2o was obtained in 87% yield while its C(sp³)-Cl bond remained untouched, again suggesting that HAT hydrogenation outcompeted halide abstraction.

Comments 3-5 and 3-7:

“There is only one example of a hydrophobic substrate (2q). Are there more examples which work? I think this is where solvents can play a role.”

“The solvent combination is a potential limitation for substrates which are hydrophobic.”

Our response:

Our attempts to employ less polar solvent systems has not been successful so far (see Table S2 in Supplementary Information). However, additional study on hydrogenation of nonpolar alkenes revealed that the reaction proceeded in synthetically useful yield (65% to 84%). These results indicate that the present system could also be useful for hydrophobic substrates. The outcomes of the reactions were added in revised Table 2 and explanation for the hydrogenation of nonpolar substrates were added in the main text.

Added substrates in revised Table 2:

Added text:

The aqueous solvent system required for this catalytic hydrogenation should be partly responsible for low yield of **2r**. Nevertheless, hydrogenation of alkenes without polar functional groups proceeded in synthetically useful yield (**2s**, **2t**, **2u**).

Comment 3-6:

“Is C-functionalized acetylated glucose (or similar substrate) a competent substrate for the reaction?”

Our response:

The C-functionalized acetylated galactose was found to be a competent substrate and was hydrogenated to afford **2ag** in 92% yield. This result was added in revised Table 2 and was explained in the main text.

Added substrates in revised Table 2:

Added text:

The catalytic hydrogenation was readily applicable to C-glycosides derived from ribofuranose (**2af**) and galactose (**2ag**).

[Additional revisions]

Additional revision 1:

Satoshi Maeda was added as a new coauthor. He contributed to quantum chemical calculations and preparation of the revised manuscript.

Revised list of authors:

Yuji Kamei,¹ Yusuke Seino,¹ Yuto Yamaguchi,¹ Tatsuhiko Yoshino,¹ Satoshi Maeda,^{2,3,4} Masahiro Kojima,^{1*} and Shigeki Matsunaga^{1,5*}

¹Faculty of Pharmaceutical Sciences, Hokkaido University, Sapporo 060-0812, Japan

²Institute for Chemical Reaction Design and Discovery (WPI-ICReDD), Hokkaido University, Sapporo 001-0021, Japan

³Faculty of Science, Hokkaido University, Sapporo 060-0810, Japan

⁴JST, ERATO Maeda Artificial Intelligence for Chemical Reaction Design and Discovery

Project, Sapporo 060-0810, Japan

⁵Global Station for Biosurfaces and Drug Discovery, Hokkaido University, Sapporo 060-0812, Japan

Additional revision 2:

A new text was added at the end of the revised abstract in order to claim that the proposed mechanism was supported by experimental studies and quantum chemical calculations.

Added text in the revised abstract:

The proposed mechanism was supported by experimental and theoretical studies.

Additional revision 3:

Recent work by Kattamuri and West was newly cited as ref 37 and mentioned in the main text as a relevant example of HAT hydrogenation.

Added reference:

37 Kattamuri, P. V. & West, J. G. Hydrogenation of alkenes via cooperative hydrogen atom transfer. *J. Am. Chem. Soc.* (2020) DOI:10.1021/jacs.0c09544.

Added text:

During the review of this manuscript, Kattamuri and West reported iron and thiol-cocatalyzed oxidant-free HAT hydrogenation of alkenes.³⁷ Nonetheless, stoichiometric silane is required as a reductant in their catalytic system.

Additional revision 4:

Acknowledgements to the followings were added.

- (1) Research grants to S.Maeda
- (2) Research grants to common HRMS facilities
- (3) WPI-ICReDD (for educational program to M.K. and computational resources)
- (4) Dr. Takakura and Prof. Dr. Ogawa (for support in photochemical studies)

Revised acknowledgements:

This work was supported in part by JSPS KAKENHI grant Number JP17H03049 and JP20H02730 (to S.Matsunaga), JSPS KAKENHI grant Number JP19K21218 and JP20K15946 (to M.K.), JSPS-WPI and JST-ERATO (No. JPMJER1903) (to S.Maeda), as well as Platform Project for Supporting Drug Discovery and Life Science Research (BINDS) from AMED under Grant No. JP20am0101093 (to common HRMS facilities). M.K. learned and performed quantum chemical calculations at the Institute for Chemical Reaction Design and Discovery (ICReDD), Hokkaido University, which was established by World Premier International Research Initiative (WPI), MEXT, Japan. We thank Dr. Kenichi Matsuda and Prof. Dr. Toshiyuki Wakimoto in Hokkaido University for allowing access to their lyophilization machine. We thank Dr. Hideo Takakura and Prof. Dr. Mikako Ogawa in Hokkaido University for their support in the Stern-Volmer quenching studies. We also thank Dr. Takuro Suzuki in our group for initially providing **3d**. Y.K. thanks the Kawamura Scholarship Foundation for a fellowship.

Additional revision 5:

Author contributions were revised considering the addition of theoretical study.

Revised author contributions:

Y.K., Y.S., Y.Y. and M.K. performed the experiments and analyzed the data. S.Maeda

and M.K. performed the quantum chemical calculations. Y.K., T.Y., S.Maeda, M.K. and S.Matsunaga conceived and designed the experiments and prepared the manuscript. All authors contributed to discussions and commented on the manuscript.

Additional revision 6:

Reference 20, which had been an *Early View* article at the time of submission, was updated.

Updated Reference 20:

20 Matos, J. L. M. *et al.* Cycloisomerization of olefins in water. *Angew. Chem. Int. Ed.* **59**, 12998-13003 (2020).

Additional revision 7:

Former reference 34 (Timmerman, J. C. *et al.* *J. Am. Chem. Soc.* **141**, 10082-10090 (2019)), which had been cited as one of the recent examples of total synthesis using HAT hydrogenation, was removed in the revised manuscript so that the total number of references does not exceed 70.

REVIEWERS' COMMENTS

Reviewer #1 (Remarks to the Author):

The authors have fully addressed my questions and comments. I appreciate it. Publication as it is is recommended.

Reviewer #2 (Remarks to the Author):

The Authors have conducted an impressive work during the revision of the present ms and have addresses satisfactorily almost all the issues raised by the Reviewers.

Acceptance in the present form is strongly recommended!

Reviewer #3 (Remarks to the Author):

This revised manuscript describes chemoselective hydrogen atom transfer (HAT) to unactivated alkenes using photoredox/Co catalysis. The authors have adequately addressed issues raised by the three reviewers during its initial submission. They have carried several more experiments which include control experiments as well as expanded the substrate scope. Computational analysis of the transition has also been included in the revised manuscript. Overall, the manuscript is well written and has important results of broad interest and provides a solution to existing methods which have shortcomings.

I recommend the manuscript in the current form for publication with minor corrections.

Table 1: Y (mol%) under column heading

Include a short description of the method used for the computational studies in the main text.

List of Revisions and Responses to Reviewer #3

[Responses to Reviewer #3]

This revised manuscript describes chemoselective hydrogen atom transfer (HAT) to unactivated alkenes using photoredox/Co catalysis. The authors have adequately addressed issues raised by the three reviewers during its initial submission. They have carried several more experiments which include control experiments as well as expanded the substrate scope. Computational analysis of the transition has also been included in the revised manuscript. Overall, the manuscript is well written and has important results of broad interest and provides a solution to existing methods which have shortcomings.

I recommend the manuscript in the current form for publication with minor corrections.

Comment 3-1:

Table 1: Y (mol%) under column heading

Our response:

Table 1 is revised accordingly.

Table 1. Evaluation of the reaction conditions for the combined cobalt/photoredox-catalyzed hydrogenation of 1a ^a			
			
Entry	Co complex (X mol%)	Y (mol%)	Yield (%) ^b
1	CoCl ₂ ·6H ₂ O (5.0)	1.0	0
2	Co(acac) ₂ ·2H ₂ O (5.0)	1.0	0

Part of revised Table 1:

Comment 3-2:

Include a short description of the method used for the computational studies in the main text.

Our response:

The method used in computational studies is now described in the main text.

Revised text:

The calculated pathway using $U\omega B97X-D$ density functional [PCM (2-Propanol), LANL2DZ(f) for Co, 6-31G(d,p)] revealed that the barrier for the exergonic HAT between the two starting materials (SM) is only 7.3 kcal/mol higher in free energy relative to the SM at 25 °C (Figure 5a), supporting that the HAT should proceed rapidly under the current reaction conditions.

Additional Revisions in Response to Editorial Requests

Additional revision 1:

The chemical structures in Figures 1,2,4,6 and Table 1,2,3 are fixed to the requested chemdraw format.

Additional revision 2:

The abstract is modified so that the current work is described in present tense.

Additional revision 3:

Legends for Figures 1,2,3,4,5,6 and Table 2 are added or modified.

Additional revision 4:

Unnecessary italics in Figures 1,2 and bold in Figure 5a are corrected.

Additional revision 5:

Reference 37 is updated as follows.

37 Kattamuri, P. V. & West, J. G. Hydrogenation of alkenes via cooperative hydrogen atom transfer. *J. Am. Chem. Soc.* **142**, 19316-19326 (2020).